# DEBIAS YOUR VLM WITH COUNTERFACTUALS: A UNIFIED APPROACH

## ABSTRACT

Recent advances in vision-language research have produced numerous foundation models that excel in tasks such as image classification, image-text retrieval, and image captioning. However, these models are shown to exploit spurious correlations in biased training data, raising fairness concerns for discrimination against underprivileged groups. In this work, we propose **CVLD**, a unified framework for quantifying and mitigating vision-language biases in a task and domain-agnostic setting. By defining a causal intervention module that produces counterfactual image-text pairs, we apply causal fairness metrics to capture the discrepancy between model predictions on original and counterfactual distributions. Building on the universal fairness notion, we propose a set of bias-free adaptation techniques to mitigate the bias of pre-trained VL models by optimizing their robustness to interventions on the protected attribute, requiring minimal modification to the naive training pipeline. **CVLD** demonstrates robust debiasing results on image classification, retrieval and captioning using adaptation datasets of varying sizes, validating the importance of counterfactual data in studying vision-language bias.

## 1 INTRODUCTION

Vision-language models (VLM) have undergone a dramatic evolution in recent years, driven by the development in cross-modal architectures, large-scale image-text corpora and pretraining techniques such as masked modeling and contrastive learning. A prominent trend in this evolution is the transition from *specialized* models, such as those for image captioning (Karpathy & Fei-Fei, 2015; Vinyals et al., 2015; Rennie et al., 2017), to *foundation* models that learn generic cross-modal representations for a broader spectrum of tasks Chen et al. (2020); Li et al. (2021); Singh et al. (2022); Alayrac et al. (2022). By pre-training a base network on massive amounts of paired image and text, foundation models are capable of learning basic visual and textual concepts as well as their correspondences, applying them to solve downstream tasks with minimal adaptation effort.

Since their introduction, concerns have arisen about the fairness and potential biases of VLMs. However, the existing work on vision-language bias has primarily targeted specialized models in their respective disciplines (Hendricks et al., 2018; Wang et al., 2021; Berg et al., 2022; Cho et al., 2022), as little effort was made towards unifying the study of biases across vision-language problems. In the vision that tasks like image classification, captioning and retrieval are to be addressed with a shared base architecture, a universal notion of bias is indispensable for the holistic understanding of biases in the emerging VLMs. However, such studies are difficult in practice as biases are often subtle and task-dependent, and mitigating them requires careful modular design tailored to each task.

Inspired by existing work on counterfactual fairness (Kusner et al., 2017) in machine learning, we argue that the key to unifying biases in VL is the ability to generate *counterfactual* data, i.e., images and text that are consistent with the true data distribution but differ in certain aspects from the original examples. This is because all VL tasks can be viewed as abstract functions of image and/or text input, and counterfactuals can be used to evaluate the robustness of such mappings to changes in the input. For example, gender bias in VLMs can be studied by swapping the gender of subjects in the input image or text, without prior knowledge of the downstream task.

However, generating counterfactual data requires the ability to perform *interventions* to particular attributes of the input, which is challenging given the lack of explicit causal models for images and text. This work investigates the use of high-quality image *editing* models to address these challenges. Following the success of diffusion models (Ho et al., 2020; Rombach et al., 2022) for

Figure 1: **CVLD** framework for debiasing vision-language models with counterfactual image-text editing.

image generation, recent research has led to methods for editing images using text prompts (Mokady et al., 2023; Brooks et al., 2023), which can be applied to real images to alter certain visual attributes of subjects, while preserving the overall appearance of the image.

In this paper, we propose a simple debiasing framework, *counterfactual vision-language debiasing* (**CVLD**), that demonstrates striking effectiveness for the most studied problems in the bias literature, including image classification, text-to-image retrieval, and image captioning. In essence, **CVLD** consists of three steps: *prompt generation* that produces instance-based text guidance for image editing, *counterfactual generation* that intervenes input images with prompted editing models, and *debiased tuning* that adapts a pre-trained model in counterfactual examples to remove unwanted biases. We show that under **CVLD**, debiasing a novel VL task reduces to a simple regularized training objective and minimal modifications to the adaptation pipeline.

Figure 1 provides an overview of the proposed framework and its advantage over existing methods for applying VLMs: 1) pre-trained VLMs usually exhibit decent fairness due to the volume and diversity of training data, but the performance of zero-shot models may be insufficient for the specific tasks and domains. 2) Naïve adaptation with a small training set greatly improves the performance of the VLM, but causes it to overfit to the specific biases of the target dataset, sacrificing fairness. 3) **CVLD** allows for unbiased adaptation by augmenting the training set with counterfactual examples, providing the best trade-off between performance and fairness.

The proficiency of **CVLD** is demonstrated in a set of fine-tuning experiments on image classification, retrieval and captioning, using well-established fairness measures. We show that **CVLD** can effectively mitigate biases in downstream tasks, while maintaining or even improving the performance of the VLMs, even on a small adaptation dataset (e.g., 1% COCO). Ablation studies and qualitative analysis further validate the importance of counterfactual data in mitigating vision-language bias.

## 2 RELATED WORKS

**Vision-language models.** Early works on VLMs focused on specialized models for specific tasks, such as image captioning (Vinyals et al., 2015; Rennie et al., 2017) and image-text matching (Ma et al., 2015; Lee et al., 2018). More recently, a wave of vision-language *foundation* models has emerged (Su et al., 2019; Li et al., 2020; Chen et al., 2020; Li et al., 2021; Singh et al., 2022; Alayrac et al., 2022; Li et al., 2022; 2023). These models are equipped with more versatile transformer architectures, pre-trained with massive amounts of image-text data, and capable of adapting to a broad spectrum of downstream tasks with little adaptation effort. The development of foundation VLMs also led to evolution in text-to-image generation (Rombach et al., 2022) and editing (Brooks et al., 2023), since state-of-the-art models are now capable of producing high-quality images consistent with text prompts. Recent work has demonstrated the potential to use these synthetic images to train strong classification models (Sariyildiz et al., 2023).

**Bias in vision-language.** Despite the long history of research on fairness and bias in machine learning (Zafar et al., 2017; Hardt et al., 2016), computer vision (Torralba & Efros, 2011; Wang et al., 2020) and natural language processing (Bolukbasi et al., 2016; Caliskan et al., 2017), only in recent years have researchers started to investigate the bias of multimodal tasks. Similarly to the path of VLMs, early work on vision-language bias focused on specialized models, revealing unwanted associations and discriminations in image captioning (Hendricks et al., 2018; Hirota et al., 2022), image-text retrieval (Wang et al., 2021; Berg et al., 2022) and text-to-image generation (Cho et al., 2022; Luccioni et al., 2023). Unlike these works, we focus on a task-agnostic fairness framework for VLMs, unifying the study of bias across different tasks and domains.

**Bias mitigation with synthetic data.** Most existing work on mitigating VLM bias employs *model* debiasing, introducing task-specific modifications to model architectures or training objec-

tives (Chuang et al., 2023; Seth et al., 2023; Hirota et al., 2023). A less popular option studied in vision models is *dataset* debiasing, by removing unwanted biases from the training data through resampling (Li & Vasconcelos, 2019) or image synthesis (Ramaswamy et al., 2021). This work follows the same direction, but extends to a multimodal setup and focuses on the use of image editing models for counterfactual generation. Closely related to this paper is the concurrent work by Smith et al. (2023), which uses image editing to create a synthetic test set for evaluating the fairness of VLMs; in contrast, we focus on the use of counterfactuals for debiased model adaptation.

# 3 IMAGE-TEXT INTERVENTION FOR HOLISTIC BIAS ESTIMATION

In this section, we present the fundamentals of fairness measures in machine learning and introduce a new framework for generating counterfactual examples for image-text data.

## 3.1 FAIRNESS IN MACHINE LEARNING

The fairness and bias of machine learning models have been extensively studied in the literature, leading to a wide spectrum of bias measures to assess trained models. Most widely adopted are *group fairness* measures, such as demographic parity (Zafar et al., 2017) and equal opportunities (Hardt et al., 2016), aiming to achieve similar prediction outcome (detection rate, precision, etc.) for different demographic groups. These are typically evaluated on *real* data distributions, thus requiring a manually balanced test set, which can be tricky to obtain for datasets collected in the wild.

*Counterfactual fairness* (Kusner et al., 2017), on the other hand, is defined on an individual basis, requiring robust system output when protected attributes (e.g., gender or race) of the inputs are altered. Formally, given a causal graph $\mathcal{G}$ of latent variables $U$, protected attributes $A \in \mathcal{A}$, other observable features $X$, and target variable $Y$, a *counterfactually fair* predictor $\hat{Y}$ of $Y$ satisfies

$$P(\hat{Y}_{A \leftarrow a}(U) = y \mid X = x, A = a) = P(\hat{Y}_{A \leftarrow a'}(U) = y \mid X = x, A = a), \tag{1}$$

where $\hat{Y}_{A \leftarrow a'}(U)$ or $\hat{Y}_{a'}$ in short is the output when an *intervention* is performed on $A$ by substituting its value with $a'$, without altering any of its non-descendants in graph $\mathcal{G}$ (i.e., features not caused by $A$). We next show how to apply counterfactual fairness to vision-language problems by recovering the causal graph $\mathcal{G}$ of image-text data and performing interventions on the protected attributes.

## 3.2 CAUSAL IMAGE-TEXT INTERVENTION

Counterfactual fairness is desirable in many real-world applications, as it demands that the model predictions be invariant to the protected attribute. This is particularly vital for the emerging VLMs trained on uncurated web datasets, which are known to exhibit biases against certain demographic groups, such as the association between gender and occupation (Seth et al., 2023). Instilling counterfactual fairness in such models would ensure that protected attributes like gender or race do not affect the model predictions, thus mitigating the risk of discrimination against underprivileged groups.

However, the counterfactual fairness measure is intractable in real-world vision-language problems, as the intervention requires access to the ground-truth causal graph $\mathcal{G}$ of the data distribution, which is impossible to recover a posteriori for large-scale datasets sourced from the Internet. Instead, we consider using *language models* and *text-to-image editing* models as a surrogate for $\mathcal{G}$.

Consider a causal graph for image or text data illustrated in Fig. 2a. Let $U$ be some latent background variables that capture the nature of a scene, which influences both the protected attributes $A$ (e.g., gender or race of people in the scene) and other semantic attributes $X$ (e.g., occupation, activity, or background objects). A visual or textual depiction of the scene $I$ can then be generated from $A$ and $X$. Given $I$ as input to a generalized[1] vision-language model $\hat{Y} = g(I)$, evaluating counterfactual fairness w.r.t. $A$ reduces to learning the counterfactual distribution of $I$:

$$P(\hat{Y}_{a'} \mid I = i, A = a) = P(g(I_{a'}) \mid I = i, A = a). \tag{2}$$

Assuming that the input $I$ is determined only by $X$ and $A$, the intervention $I_{a'}$ can be generated by substituting $A$ with $a'$ in $\mathcal{G}$ while keeping $X$ unchanged. This is straightforward for text inputs, e.g., by replacing all gendered words in the text, thanks to the high abstraction of the modality, which

---

[1]Here we consider a generalized system that accepts inputs from either modalities, which encompasses most existing VL models including image captioning, text-to-image retrieval, and visual question answering.

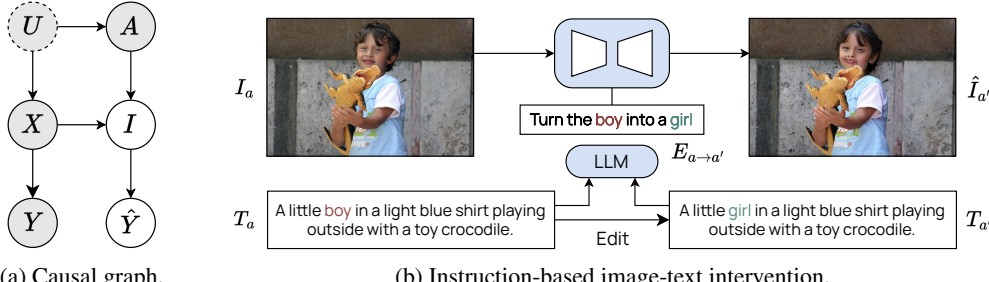

(a) Causal graph.  (b) Instruction-based image-text intervention.

Figure 2: Protected attribute intervention with text-guided image editing.

naturally disentangles $A$ from $X$.[2] Yet for images, a similar intervention is nontrivial, requiring a generative model that can perform targeted manipulations on $A$ while preserving other visual attributes $X$.

With recent advances in text-guided image editing (Hertz et al., 2022; Mokady et al., 2023; Brooks et al., 2023), we argue that it is indeed possible to perform such interventions on real images. This is due to two key observations: First, prompt-to-prompt (P2P) editing (Hertz et al., 2022) allows precise image manipulation by swapping cross-attention maps between the original and edited prompts. This makes it possible to perform *local* edits to the protected attribute of interest ($A$) without changing the overall image structure and contextual information ($X$). Second, inversion-based image editing (Mokady et al., 2023) produces accurate attention grounding from text prompts to the image regions, which can be used to guide the editing process. Likewise, instruction-based editing models (Brooks et al., 2023) trained on P2P-generated data also inherit the ability to perform local edits, while being more robust to complex scenes.

### 3.3 INTERVENTION PROCEDURE

Having established the feasibility of image-text intervention, we next describe the procedure of generating counterfactual examples for vision-language data. As illustrated in Fig. 2b, the intervention procedure consists of three steps: *text intervention*, *prompt generation*, and *image editing*.

**Text intervention.** Given an dataset of paired image-text examples with protected attribute annotations $\mathcal{D} = \{(I, T, A)\}$, we first perform text intervention on the captions $T$ to generate counterfactual examples $T_{a'}$ with protected attributes $A$ substituted for $a'$. While a naïve rule-based method is possible, it neglects the overall structure of the sentence, causing grammatical errors and semantic inconsistencies especially when more than one subjects are involved (e.g., "a boy and a girl" becomes "a boy and a boy"). Instead, we use the GPT-3.5 (Brown et al., 2020) large language model to process $T$, which produces more natural and fluent counterfactual examples.

**Prompt generation.** Given the source text $T$ and target $T_{a'}$, an *inversion*-based pipeline, e.g., null-text inversion (Mokady et al., 2023) with prompt-to-prompt editing (Hertz et al., 2022), can be used to generate the corresponding counterfactual images $I_{a'}$ directly. Alternatively, one can use an *instruction*-tuned model like InstructPix2Pix (Brooks et al., 2023) by providing with the image a text prompt that describes the edit (e.g., "turn the boy into a girl"). As with text intervention, we use GPT-3.5 to generate text prompts for InstructPix2Pix using both $T$ and $T_{a'}$ as input.

**Image editing.** Once the source/target text and edit prompts are generated, we can perform image editing to produce the counterfactual images $I_{a'}$ using null-text inversion or InstructPix2Pix. Notably, even with state-of-the-art editing models, the edited images may not be semantically consistent with the text prompts or deviate substantially from the input image layout. To mitigate this variance, we repeat the editing $m$ times with each method, and introduce a filtering step that selects the $k < m$ top candidates with the highest similarity scores to the original image, measured by the cosine similarity of their feature embeddings from a pre-trained image encoder.

**Image-only intervention.** While the above procedure is designed for image-text data, it can be easily adapted to image-only data by using manually designed prompts. For example, we can use "a photo of a man/woman" in null-text inversion, and "turn the man/woman into a woman/man" in InstructPix2Pix. Although more sophisticated strategies are possible (e.g., using a captioning model), we use the simple method described above, as it is sufficient for our experiments.

---

[2]This is a simplified view of text intervention, as some words like "actor/actress" implies both gender and occupation. However, this can be easily addressed by using more sophisticated language models.

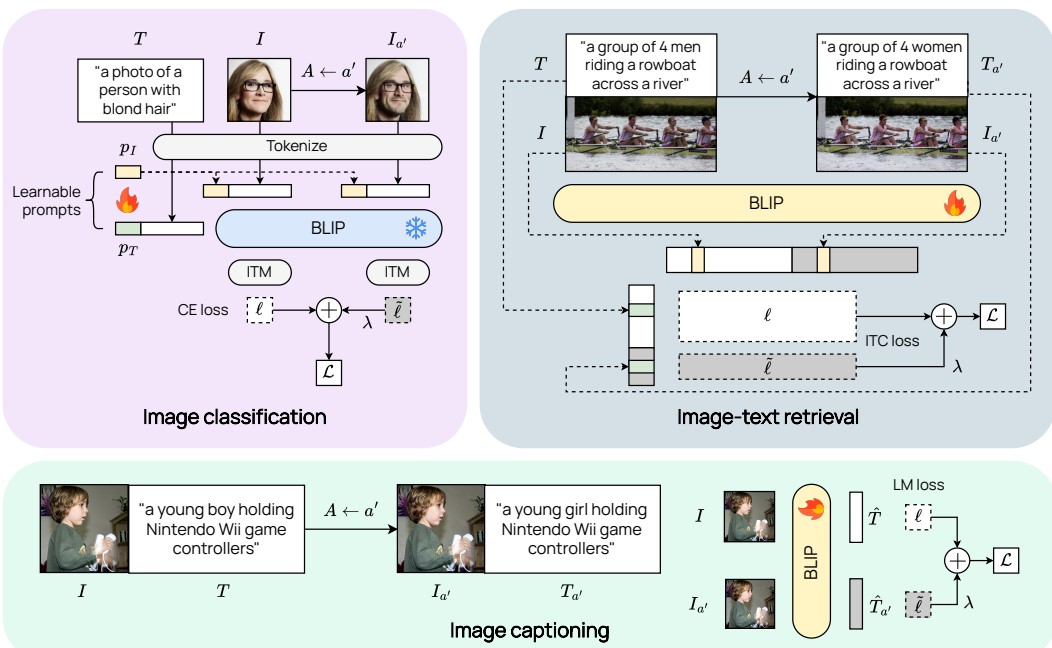

Figure 3: Bias-free adaptation of BLIP on different VL tasks. **CVLD** enables the debiasing of foundation VLMs on classification, retrieval and captioning tasks, with minimal modification to the training pipeline.

# 4  **CVLD**: BIAS-FREE ADAPTATION OF FOUNDATION MODELS

Successful application of foundation VLM often requires *adaptation* to a small number of examples from the target distribution, which can cause it to overfit to dataset-specific biases. In this section, we introduce a novel method for unbiased adaptation of VLMs to image classification (Sec. 4.2), retrieval (Sec. 4.3) and captioning (Sec. 4.4) tasks that facilitates synthetic counterfactuals of Sec. 3.

## 4.1  PRELIMINARIES

**Architecture.**   We use BLIP (Li et al., 2022) as the base model for bias-free adaptation. BLIP is a versatile VLM containing a ViT (Dosovitskiy et al., 2020) image encoder and a BERT (Devlin et al., 2018) text encoder modified with additional cross-modal attention layers, demonstrating strong performance on various vision-language tasks. Its components and training are also representative of other foundation VLMs such as Florence (Yuan et al., 2021) and BLIP-2 (Li et al., 2023).

**Training with counterfactuals.**   The key intuition of **CVLD** is to augment the training set $\mathcal{D} = \{I\}$ with counterfactual examples $\tilde{\mathcal{D}} = \{I_{a'} \mid a' \in \mathcal{A}, I \in \mathcal{D}\}$, and optimize the model on the combined set $\mathcal{D} \cup \tilde{\mathcal{D}}$. In the case of binary protected attributes such as gender, it suffices to partition the training set into subgroups $(\mathcal{D}_1, \mathcal{D}_{-1})$ containing examples of attribute $A = 1$ or $-1$, and construct counterfactuals by flipping the attribute of each example in $\mathcal{D}_a$, i.e. $\tilde{\mathcal{D}}_a = \{I_{-a} \mid I \in \mathcal{D}_a\}$.

Finally, given a batch of training examples $\mathcal{B} = \{(I, T)\}$ and their counterfactuals $\tilde{\mathcal{B}} = \{(\tilde{I}, \tilde{T})\}$, the augmented loss for model $\theta$ is defined as

$$\mathcal{L}_\lambda(\theta; \mathcal{B}, \tilde{\mathcal{B}}) = \ell(\theta; \mathcal{B}) + \lambda \ell(\theta; \tilde{\mathcal{B}}), \tag{3}$$

where $\lambda$ is a hyperparameter for the trade-off between the original and counterfactual losses. In the following sections we will elaborate on the task-specific implementations of the **CVLD** framework.

## 4.2  IMAGE CLASSIFICATION

Foundation VLMs like CLIP (Radford et al., 2021) pre-trained on massive image-text corpora have shown remarkable zero-shot classification capabilities, simply by evaluating the cross-modal affinity between input images and text prompts $p(c)$ of each class $c$ (e.g., $p(c)$ = "a photo of a [c]"):

$$f(I, \mathcal{C}; \theta) = \arg\max_{c \in \mathcal{C}} g(I, p(c); \theta). \tag{4}$$

The affinity score $g(I, T; \theta)$ can be implemented by cosine similarity in the joint embedding space (as in CLIP), or in the case of BLIP a dedicated image-text matching (ITM) head trained with binary classification loss. When zero-shot classification produces suboptimal results, fine-tuning is needed on a small set of examples to adapt the VLM to the target distribution. However, adaptation may also inject biases specific to the target dataset into the foundation model.

As illustrated in Fig. 3 (top left), we use a multimodal prompt tuning strategy (Jia et al., 2022; Zang et al., 2022; Khattak et al., 2023) to perform BLIP adaptation while using counterfactual examples to mitigate model bias. Specifically, we create learnable prompt embeddings $\boldsymbol{p}_I$ and $\boldsymbol{p}_T$ and prepend them to the tokenized inputs to the visual and text transformer encoders, respectively. During few-shot adaptation, only the prompt embeddings of both modalities are updated, while the rest of the model parameters are frozen. The image and text prompts are updated with gradient descent on the augmented loss of Eq. (3), where $\ell(\theta; \mathcal{B})$ denotes the standard cross-entropy classification loss:

$$\boldsymbol{p}_I \leftarrow \boldsymbol{p}_I - \eta \nabla_{\boldsymbol{p}_I} \mathcal{L}_\lambda(\theta; \mathcal{B}, \tilde{\mathcal{B}}), \quad \boldsymbol{p}_T \leftarrow \boldsymbol{p}_T - \eta \nabla_{\boldsymbol{p}_T} \mathcal{L}_\lambda(\theta; \mathcal{B}, \tilde{\mathcal{B}}). \tag{5}$$

Note that on a labeled classification dataset $\mathcal{D} = \{(I, C, A)\}$ with images $I$, ground-truth classes $C$ and protected attribute $A$, applying equation 3 with $\lambda = 1$ effectively decorrelates the $C$ with $A$:

$$p_{C,A}(\mathcal{D} \cup \tilde{\mathcal{D}}) = p_{C,A}(\mathcal{D}) + p_{C,A}(\tilde{\mathcal{D}}) = p_{C,A}(\mathcal{D}) + p_{C,-A}(\mathcal{D}) = p_C(\mathcal{D}). \tag{6}$$

### 4.3 Image-text retrieval

Image retrieval with BLIP is a two-stage procedure following earlier work of Li et al. (2021). First, $K$ image nearest neighbors to the text query are identified in their joint embedding space. This is trained using an image-text contrastive (ITC) loss[3] based on InfoNCE (Oord et al., 2018):

$$\ell_{\text{ITC}}(\theta; \mathcal{B}, \mathcal{B}^-) = \mathbb{E}_{(I,T) \in \mathcal{B}} \left[ -\log \frac{\exp \langle g_I(I; \theta), g_T(T; \theta) \rangle}{\sum_{(I',*) \in \mathcal{B}^-} \exp \langle g_I(I'; \theta), g_T(T; \theta) \rangle} \right]. \tag{7}$$

The likelihood of each nearest-neighbor candidate is then evaluated with the binary ITM head, the output of which is used to refine the initial retrieval rankings. The ITM loss is a standard binary cross-entropy loss, with hard negative sampling by upweighting negative pairs with high ITC scores.

To see how this retrieval framework can be debiased by **CVLD**, consider a batch of training examples $\mathcal{B} = \{(I, T)\}_{i=1}^N$ consisting of paired images $I_i$ and captions $T_i$. Not all of these examples contain information about the protected attribute $A$; for the subset of examples $\mathcal{B}_a \subset \mathcal{B}$ that do, a counterfactual set can be constructed with flipped attribute, i.e., $\tilde{\mathcal{B}}_a = \{(I_{-a}, T_{-a}) \mid (I, T) \in \mathcal{B}_a\}$.

**Candidate extension.** An abstract illustration of **CVLD** for debiasing ITC loss is shown in Fig. 3 (top right). In retrieval, both the query set (text prompts) and candidate set (images) can be augmented with counterfactuals. First, we expand the set of candidates by including the batch of intervened images $\{I \mid (I, T) \in \tilde{\mathcal{B}}_a\}$. Instead of treating these images as separate instances (negatives to original queries), we use the *unimodal* similarity between the original and intervened *text* queries $T$ and $T_{-a}$ to supervise ITC learning. This makes sense as for a neutral prompt $T$ (e.g., "a photo of a person"), both $I$ and $I_{-a}$ should be considered positives despite having opposite gender attributes.

**Query extension.** To further improve the fairness of retrieval, we introduce counterfactual text queries $\{T \mid (I, T) \in \tilde{\mathcal{B}}_a\}$ and compute the augmented ITC loss using Eq. (3):

$$\mathcal{L}(\theta; \mathcal{B}, \tilde{\mathcal{B}}) = \ell_{\text{ITC}}(\theta; \mathcal{B}, \mathcal{B} \cup \tilde{\mathcal{B}}) + \lambda \ell_{\text{ITC}}(\theta; \tilde{\mathcal{B}}, \mathcal{B} \cup \tilde{\mathcal{B}}). \tag{8}$$

The debiasing for ITM also follows the formulation of Eq. (3), although we exclude counterfactual examples from being chosen as hard negatives, for similar reasons to the use of soft labels in ITC above.

### 4.4 Image captioning

Figure 3 (bottom) shows the pipeline for debiasing image captioning with **CVLD**. It follows a similar procedure to classification (Sec. 4.2), except that the ground-truth captions $T$ are also intervened to match counterfactual images $I_{a'}$, and the classification loss is replaced with language modeling loss:

$$\ell(\theta; \mathcal{B}) = \mathbb{E}_{(I,T) \in \mathcal{B}} \left[ -\log p(T \mid I; \theta) \right]. \tag{9}$$

Also, as with image classification, counterfactual augmentation decorrelates the protected attribute $A$ with all other words in the text, thus removing spurious associations in the original dataset.

---

[3]BLIP also uses a momentum encoder to augment positive and negative sets, omitted here for simplicity.

|  | **Fairness** | | **Performance** | |
| --- | --- | --- | --- | --- |
|  | DEO ↓ | ΔAcc ↓ | AP ↑ | mAcc ↑ |
| BLIP[PT] | **10.4** | 46.2 | 54.4 | 66.1 |
| BLIP | 14.2 | **27.6** | **85.2** | **84.2** |
| RN-50 (He et al., 2016) | 16.7 | – | 83.9 | – |
| +Ramaswamy et al. (2021) | 13.9 | – | 83.0 | – |
| *256-shot* | | | | |
| BLIP | 16.0 | 14.2 | 88.7 | 82.8 |
| **CVLD** | **10.2** | **11.2** | **89.4** | **83.9** |
| *16-shot* | | | | |
| BLIP | 20.1 | 20.7 | **82.3** | 76.5 |
| **CVLD** | **15.6** | **18.2** | 82.2 | 76.5 |

Table 1: Image classification on CelebA. All metrics are computed over 14 gender-independent target attributes.

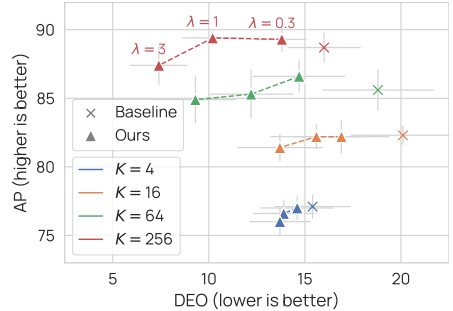

Figure 4: $K$-shot image classification on CelebA. Compared to naïve cross-entropy training, **CVLD** greatly reduces bias with little performance loss.

## 5 RESULTS

In this section, we evaluate the effectiveness of **CVLD** in debiasing vision-language models on a variety of downstream tasks, including image classification, image-text retrieval, and image captioning.

### 5.1 EXPERIMENTAL SETUP

We briefly describe the data and models in the experiments. Refer to the Appendix for more details.

**Datasets.** Different datasets are used to evaluate the bias of the adapted BLIP models on each downstream task. For *image classification*, we use CelebA (Liu et al., 2015) to evaluate the gender bias of BLIP in a few-shot adaptation setting. Specifically, for $K$-shot classification, $K$ images from each target class are sampled from the CelebA training set and used to optimize the pre-trained BLIP model. For *image-text retrieval*, we use COCO (Chen et al., 2015) to adapt the model and evaluate it on the test sets of COCO and Flickr30K (Plummer et al., 2015). Performance is measured by top-1 recall (Rec@1) and mean top-$K$ recall with $K \in \{1, 5, 10\}$ (mRec@$K$). We follow Berg et al. (2022) to use an explicitly balanced dataset, FairFace (Karkkainen & Joo, 2021), to evaluate gender bias in the retrieved images through Bias@$K$ and MaxSkew@$K$. For *image captioning*, we fine-tune the pre-trained BLIP model on COCO and evaluate it on both COCO and nocaps. Following prior work (Hirota et al., 2022; 2023) we evaluate the group error, bias amplification (BA) and leakage (LIC) of adapted models on COCO to quantify their fairness.

**Intervention.** We follow the procedure introduced in Sec. 3.3 to generate counterfactual examples for model adaptation. In CelebA classification, simple manual text prompts are used to flip the gender of each training image. For image-text retrieval and captioning, we use GPT-3.5 Turbo to generate counterfactual text prompts from original captions. Using either null-text inversion (Mokady et al., 2023) with prompt-to-prompt (Hertz et al., 2022) or InstructPix2Pix (Brooks et al., 2023), we generate $m = 10$ edited images per example and keep the top $k = 3$ examples with the highest cosine similarity in to the original image, using visual features of a pre-trained Swin transformer (Liu et al., 2021). The edited image-text pairs are used to augment the training set for model adaptation.

**Models.** We use the official BLIP model (Li et al., 2022) for all downstream experiments, adapting the model using the objectives detailed in Sec. 4. We use multimodal prompt tuning to adapt the model for few-shot classification and unlock the full model in retrieval and captioning tasks. Specifically, we prepend $m = 4$ learnable prompt embeddings to input sequences to each of the first 9 layers of the visual and text transformer encoders. During few-shot adaptation, only the prompt embeddings of both modalities are updated, while the rest of the model parameters are frozen.

### 5.2 DEBIASED IMAGE CLASSIFICATION

To simulate realistic scenarios of VLM adaptation, we consider a low-shot classification setup with a varying number of shots $K$ per class. For CelebA, the model is fine-tuned on $2K$ examples per attribute ($K$ positives & $K$ negatives), and compared to zero-shot classification of pre-trained BLIP.

**Results.** Table 1 compares the performance and fairness of attribute classification on CelebA using **CVLD** vs. baseline fine-tuning. We make the following findings: First, zero-shot model (BLIP[PT]) reports decent DEO bias, albeit with much lower classification performance. Its performance gap is also the largest among all models evaluated, which is due to its poor balance between positive

| | **Fairness (FairFace)** | | | | **Retrieval (COCO)** | | **Zero-shot (Flickr30K)** | |
|---|---|---|---|---|---|---|---|---|
| | Bias$^{1K}$ ↓ | MaxSkew$^{1K}$ ↓ | Bias$^{128}$ ↓ | MaxSkew$^{128}$ ↓ | Rec@1 ↑ | mRec ↑ | Rec@1 ↑ | mRec ↑ |
| BLIP$^{PT}$ | .195 | .171 | .339 | .277 | 55.0 | 73.7 | 79.7 | 90.4 |
| BLIP$_{384}$ | .328 | .271 | .437 | .350 | 63.9 | 80.5 | 85.1 | 93.5 |
| *COCO 10%* | | | | | | | | |
| BLIP | .237 | .199 | .338 | .274 | 58.5 | **76.6** | **80.7** | **90.9** |
| BLIP$^{Bal}$ | .220 | .186 | .283 | .225 | **58.7** | 76.5 | **80.7** | 90.8 |
| **CVLD** | **.172** | **.151** | **.241** | **.205** | 58.6 | 76.5 | 80.6 | 90.8 |
| BLIP$_{384}$ | .313 | .257 | .455 | .359 | **61.5** | **78.6** | 83.6 | 92.8 |
| BLIP$^{Bal}_{384}$ | .269 | **.216** | .377 | **.295** | 60.6 | 78.1 | 83.3 | 92.6 |
| **CVLD$_{384}$** | **.257** | .219 | **.367** | **.295** | 61.2 | 78.5 | **84.1** | **92.9** |
| *COCO 1%* | | | | | | | | |
| BLIP | .216 | .186 | .394 | .317 | 58.4 | 76.4 | 80.2 | 90.7 |
| BLIP$^{Bal}$ | .300 | .256 | .442 | .358 | 58.2 | 76.2 | 80.3 | 90.6 |
| **CVLD** | **.173** | **.150** | **.214** | **.179** | 58.0 | 76.1 | 80.3 | 90.7 |
| BLIP$_{384}$ | .301 | .250 | .348 | .281 | 59.7 | 77.5 | 82.1 | 92.0 |
| BLIP$^{Bal}_{384}$ | .435 | .355 | .556 | .434 | **60.1** | **77.7** | **82.5** | **92.1** |
| **CVLD$_{384}$** | **.205** | **.174** | **.270** | **.222** | 59.4 | 77.4 | 82.2 | 91.9 |

Table 2: Results on image-text retrieval. All models use 224×224 resolution unless specified in subscripts.

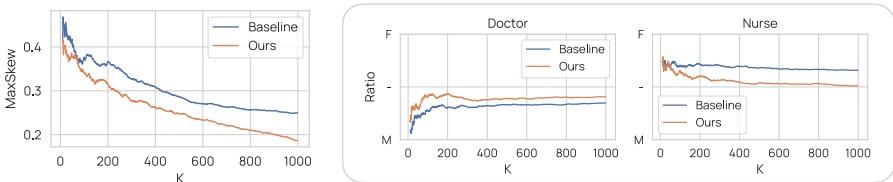

Figure 5: Qualitative results in image-text retrieval. **Left**: MaxSkew@$K$ for varying values of $K$ averaged over all text queries on FairFace; **Right**: gender ratio of top $K$ examples when querying "a photo of a doctor/nurse".

and negative predictions. Second, baseline fine-tuning with cross entropy improves classification accuracy, but also introduces significant DEO bias. This effect is even more pronounced in low-shot settings, as DEO increases from 14.2 on the full dataset to 16.0 (256-shot) and 20.1 (16-shot). Finally, **CVLD** with $\lambda = 1$ significantly reduces both DEO and $\Delta$Acc compared to fine-tuning, while maintaining or even improving the classification performance. In 256-shot classification, it produces lower biases than BLIP$^{PT}$ while attaining a classification AP more than 40% higher.

**Ablation studies.** We further vary the loss weight $\lambda$ of counterfactual examples, and plot the performance and bias of adapted models in Fig. 4. It can be seen that **CVLD** is robust to the choice of $\lambda$, with larger values producing a lower bias. We start to observe performance degradation when $\lambda > 1$ (i.e., weighting counterfactual examples over the originals) while fairness continues to improve.

## 5.3 Debiased text-to-image retrieval

Next, we apply **CVLD** to the image-text retrieval task (Sec. 4.3). We perform fine-tuning on subsets of COCO of different sizes, and report the retrieval performance and fairness of the adapted models.

**Results.** As shown in Tab. 2, naïve fine-tuning (BLIP) results in significant bias amplification compared to the pre-trained model (BLIP$^{PT}$), similar to the trend observed in image classification. In both 10% and 1% COCO, **CVLD** achieves lower bias than standard fine-tuning with comparable recall scores. Notably, alternative dataset debiasing methods such as balanced sampling (BLIP$^{Bal}$) do not produce the same level of bias reduction as **CVLD**, even *increasing* the bias in 1% COCO. This shows the importance of debiasing in low-shot settings by *augmenting* the training data, rather than resampling which effectively reduces the sample count.

| Method | FairFace MaxSkew$^{128}$ | COCO mRec |
|---|---|---|
| Base | .317 | **76.4** |
| Hard label | .416 | 76.2 |
| Neg only | .261 | 76.2 |
| ITC | .227 | 76.2 |
| ITM | .189 | 76.1 |
| **CVLD** | **.179** | 76.1 |

Table 3: Ablation on **CVLD** variants for debiasing image retrieval. All models trained on COCO 1%; 224×224 input.

**Ablation studies.** We make several observations by comparing the results of different variants of **CVLD** in Tab. 3: 1) **CVLD** with hard labels *increases* bias relative to the base model, likely because all counterfactual examples are considered negatives, forcing the model to learn gender-discriminant representations; 2) **CVLD** with counterfactuals as *negative* candidates but no additional loss ($\lambda = 0$) achieves lower bias than base BLIP, but higher than **CVLD**. This suggests that using counterfactuals

| | Fairness (COCO) | | | Captioning (COCO) | | Zero-shot (nocaps) | |
|---|---|---|---|---|---|---|---|
| | Error ↓ | BiasAmp ↓ | LIC ↓ | BLEU4 ↑ | CIDEr ↑ | CIDEr ↑ | SPICE ↑ |
| BLIP[PT] | 3.8 | 1.67 | 2.3 | 38.5 | 129.2 | 74.4 | 10.7 |
| BLIP | 4.3 | 1.34 | 4.3 | 39.6 | **133.0** | **109.5** | **14.6** |
| OSCAR (Li et al., 2020) | **3.0** | 1.78 | 2.4 | 39.4 | 119.8 | – | – |
| +LIBRA (Hirota et al., 2023) | 4.6 | −1.95 | **0.3** | 37.2 | 113.1 | – | – |
| GRIT (Nguyen et al., 2022) | 3.5 | 3.05 | 3.1 | **42.9** | 123.3 | – | – |
| +LIBRA | 4.1 | 1.57 | 0.7 | 40.5 | 116.8 | – | – |
| *COCO 10%* | | | | | | | |
| BLIP | **4.2** | 1.54 | 4.8 | 38.9 | **130.8** | 107.4 | **14.5** |
| BLIP[Bal] | 4.9 | 0.97 | 4.0 | **39.1** | 130.4 | 106.5 | 14.2 |
| **CVLD** | 4.6 | **0.12** | **1.5** | 38.5 | 129.9 | 106.8 | 14.4 |
| *COCO 1%* | | | | | | | |
| BLIP | 5.6 | 0.91 | 5.6 | 39.0 | 130.0 | 106.4 | 14.1 |
| BLIP[Bal] | **4.2** | 0.37 | 5.5 | 39.1 | 129.5 | 106.5 | 14.1 |
| **CVLD** | 5.0 | **−0.42** | **3.7** | **39.4** | **130.4** | 106.6 | **14.2** |

Table 4: Results on image captioning. All models except BLIP[PT] use 384×384 resolution.

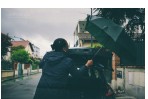

Baseline: A man holding an umbrella in the back of a car.

Ours: A woman holding an umbrella standing in the back of a car.

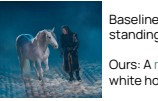

Baseline: A woman in a black dress standing next to a white horse.

Ours: A man standing next to a white horse.

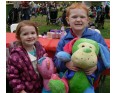

Baseline: Two little girls sitting next to each other holding stuffed animals

Ours: Two young children sitting at a table with stuffed animals.

Figure 6: Qualitative results on image captioning with baseline BLIP and **CVLD**, fine-tuned on 1% COCO.

as negatives alone is not sufficient to debias the model; 3) ours with either debiased ITC or ITM loss improves over "Neg only", showing the benefit of using counterfactuals in query sets. **CVLD** uses a combination of both, which achieves the best fairness score.

**Qualitative results.** We take a closer look at the fairness of retrieval by plotting the gender uniformity in the top $K$ retrieved images, measured by MaxSkew@$K$. As Fig. 5 (left) shows, **CVLD** produces consistently lower skew than standard fine-tuning, despite the noisy curve when $K$ is small. Beyond the standard set of queries in Berg et al. (2022), we also plot the gender ratio in retrieved images from occupation queries. As shown in Fig. 5 (right), when querying "a photo of a doctor/nurse", **CVLD** produces more balanced results, indicating a lower gender-occupation association.

## 5.4 Debiased image captioning

We finally study the debiasing performance of **CVLD** for image captioning. As fine-tuning with 224×224 resolution does not improve the scores, we use 384×384 for all adapted captioning models.

**Results.** As the results in Tab. 4 reveal, all baseline models without explicit debiasing exhibit noticeable bias amplification and leakage regardless of dataset size. This is partially remedied by balancing the training set (BLIP[Bal]), at the cost of captioning scores. A similar fairness-performance trade-off is observed in LIBRA (Hirota et al., 2023), a recent work on mitigating gender bias of image captioning models. In comparison, **CVLD** attains much lower BiasAmp and LIC scores, while maintaining similar captioning quality. When trained on 1% COCO, it produces a negative BiasAmp score, indicating a bias *reduction* vs. ground-truth annotations. As a side product of debiasing, we also observed a slight improvement in captioning performance on both COCO and nocaps, likely due to the additional intervened examples reducing overfitting of fine-tuned model.

**Qualitative results.** As Fig. 6 shows, we found that **CVLD** improves gender resolution in challenging scenes (left & center), indicating robustness to contextual bias; it also tends to produce gender-neutral captions (right), when given insufficient information to infer the gender of subjects.

## 6 Conclusion

In this work, we introduced the **CVLD** framework for debiasing vision-language foundation models using counterfactually edited image-text data. **CVLD** utilizes language models and text-guided image editing to perform causal interventions on the protected attribute, and adapts the model on the counterfactual examples without major modifications to the training pipeline. The effectiveness of **CVLD** was demonstrated on a set of vision-language tasks, including image classification, text-to-image retrieval and image captioning, where it is shown to improve the fairness of adapted VLMs with little performance loss. We hope that **CVLD** can serve as a universal framework for studying and mitigating vision-language bias, and inspire future research on the fairness of foundation models.

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

## A    DATASETS AND EVALUATION

**Image classification.**    Following the evaluation protocol in Ramaswamy et al. (2021), we use each of the 14 gender-independent binary facial attributes of CelebA (*Bags Under Eyes Bangs, Black Hair, Blond Hair, Brown Hair, Chubby, Eyeglasses, Gray Hair, High Cheekbones, Mouth Slightly Open, Narrow Eyes, Smiling, Wearing Earrings, Wearing Hat*) as target labels, and gender as the protected attribute. Classification performance is measured by average precision (AP) and mean accuracy (mAcc) over demographic groups; gender bias is evaluated by difference in equal opportunity (DEO) and accuracy gap ($\Delta$Acc). Specifically, DEO measures the difference in true positive rate between examples of each gender; $\Delta$Acc measures the difference in accuracy between the worst-performing group (defined as target attribute $Y \times$ protected attribute $A$, e.g., blond males, or female with eyeglasses) and the dataset average. The results are averaged over 10 independent few-shot episodes to account for the variance in low-shot sampling.

**Image-text retrieval.**    We evaluate text-to-image retrieval on COCO and Flickr30K using the standard procedure of BLIP (Li et al., 2022). The fairness of retrieval is evaluated on FairFace, using the standard query template by Berg et al. (2022): "a photo of a {good, evil, smart, dumb, attractive, unattractive, lawful, criminal, friendly, unfriendly} person". We use Bias@$K$ (Wang et al., 2021) and MaxSkew@$K$ (Geyik et al., 2019) to quantify the model bias, where a zero Bias@$K$ or MaxSkew@$K$ indicates a uniform representation of each gender in the top $K$ retrieved images.

**Image captioning.**    We evaluate image captioning on COCO using the standard metrics of BLEU-4, CIDEr, and on nocaps validation split using CIDEr and SPICE. The evaluation on nocaps is performed using the official evaluation server[4]. Following the protocol of (Hirota et al., 2022), the fairness of captioning is measured by group error rate (accuracy of predicting gender of the subject), bias amplification (BiasAmp; difference in gender association in ground-truth and predicted captions), and leakage (LIC; classifier accuracy in predicting gendered words in captions).

## B    IMAGE-TEXT INTERVENTION

We use the official GPT-3.5 Turbo API[5] by OpenAI to perform text intervention and prompt generation on COCO. To generate counterfactual text prompts, we use the following template:

> Copy the sentence, but follow these steps:
> - Determine if there are human in the image.
> - If no human, do not change anything.
> - If there is human, turn all female people into male. For example, replace "woman" with "man", "girl" with "boy".
> - Do not change the age of people. For example, avoid replacing "boy" with "man", or "man" with "boy".
> - Do not change anything unrelated to gender, such as ethnicity or colors.

---

[4] http://eval.ai/web/challenges/challenge-page/355/
[5] https://platform.openai.com/docs/models/gpt-3-5

To generate text prompts for image editing, we use the following template:

> Compare the gender of image captions (before–after), and find the best editing prompt such as:
> - Turn the man into a woman.
> - Turn the woman into a man.
> - Turn the boy into a girl.
> - Turn the girl into a boy.
> - Turn the woman into a man and the girl into a boy.
> - No change.

As each image in COCO is associated with multiple captions, we generate one text prompt per caption using the procedure above, and only accept the intervened example if a majority agreement is reached in the edit prompts (e.g., if 5 captions are available for image $I$, 3 of the output prompts must agree on the same edit). This leads to a smaller counterfactual dataset $\tilde{\mathcal{D}}$ (approximately 20% of the original dataset $\mathcal{D}$). Samples without counterfactuals are trained with the standard retrieval and captioning losses.

## C   TRAINING DETAILS

We follow the official BLIP implementation[6] to adapt the pre-trained models on retrieval and captioning tasks. Models are optimized for 6 epochs for retrieval and 5 epochs for captioning, using a learning rate of $10^{-5}$ and batch size of 128 over 4 GPUs. In few-shot classification, we use a batch size of 32 and a learning rate of $10^{-2}$ for multimodal prompt tuning. The loss weight $\lambda$ in `CVLD` is set to 1 for classification and 0.3 for retrieval and captioning. We intend to make the counterfactual datasets and adapted models using `CVLD` publicly available to facilitate future research on vision-language bias.

## D   QUALITATIVE RESULTS

**Edit quality.**   Figure 7 shows the original and counterfactually edited COCO examples. Although the editing process is not perfect, it successfully alters the protected attribute (gender) in most images consistent with the text edits generated by the LLM, while preserving other visual information such as background objects, posture, and hair style of the subjects. Similar findings hold for image editing in CelebA (Fig. 8), where we found `CVLD` to produce high-quality samples to minority combinations of attributes (e.g., male with earrings), by editing images from the majority group (female with earrings). This indeed translates to a significantly lower classification bias of the adapted VLM.

**Debiased image classification.**   Figure 9 shows the relative improvement in the classification bias (DEO) of `CVLD` over baseline fine-tuning on 256-shot CelebA. We find `CVLD` to be the most effective for the *most biased* attributes such as "bags under eyes" and "wearing earrings", where the original training distribution is heavily unbalanced with respect to gender.

Figure 10 shows the gender uniformity of retrieved examples for each text query on FairFace. It can be seen that `CVLD` significantly reduces gender bias for 5 of the 10 queries (*good, dumb, attractive, unattractive, friendly*), while maintaining similar levels of fairness for the rest of concepts.

## E   LIMITATIONS

While `CVLD` validates the effectiveness of text-to-image editing diffusion models for generating counterfactual vision-language data, it is not without limitations. First, the quality of counterfactual examples is bounded by the capacity of the underlying image editing model. `CVLD` relies on the capacity of state-of-the-art editing models to alter the protected attribute while keeping the remaining features unchanged, which is not guaranteed for all types of images. More importantly, the editing models may encompass their own biases, which can be transferred to the counterfactual examples and ultimately the debiased model. Further research may be needed for a comprehensive and objective evaluation of the quality of counterfactual examples. Second, the current implementation of `CVLD` is limited to debiasing downstream tasks with minimal adaptation, while the debiasing of

---

[6]https://github.com/salesforce/LAVIS

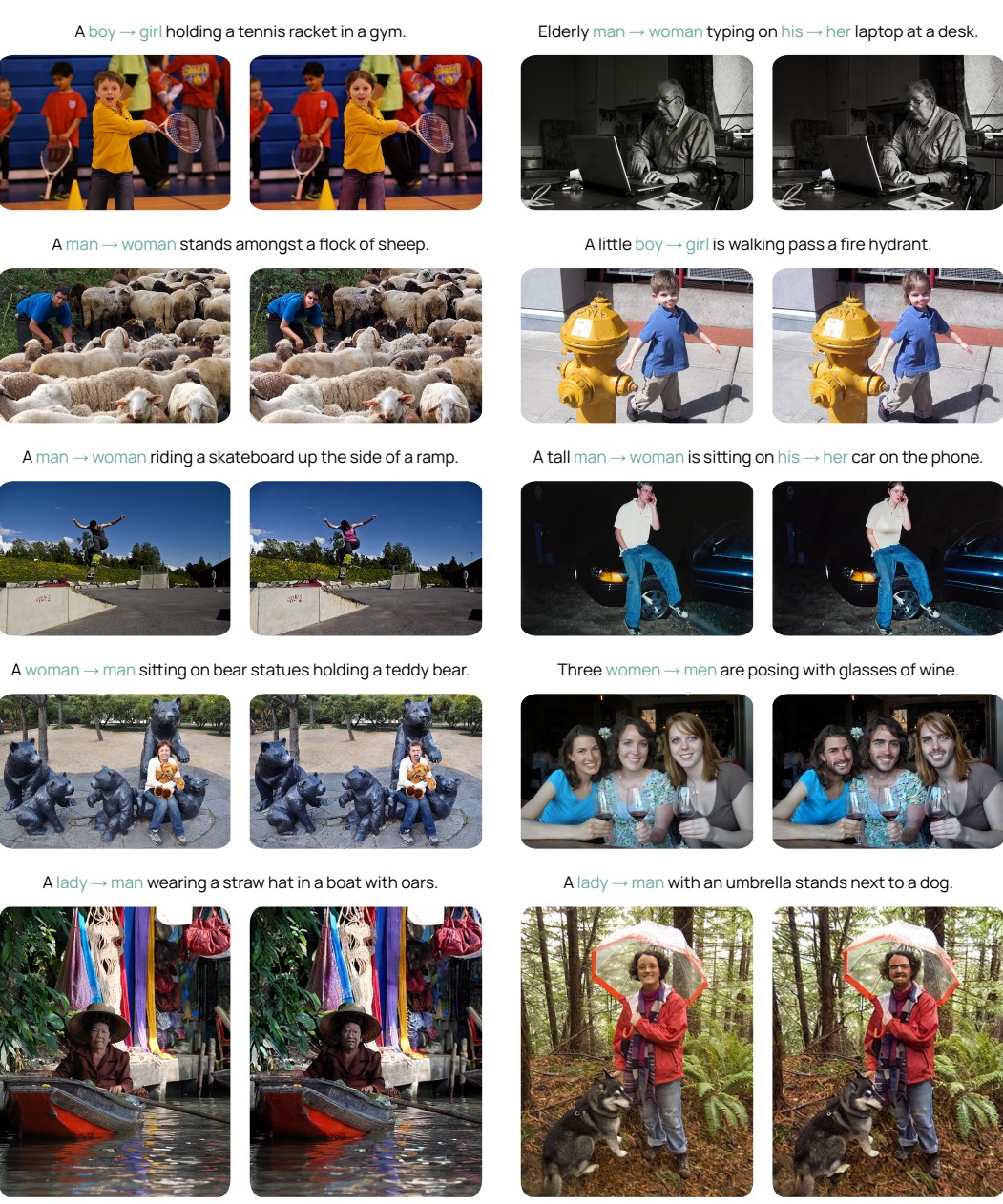

Figure 7: Counterfactual image-text editing on COCO dataset.

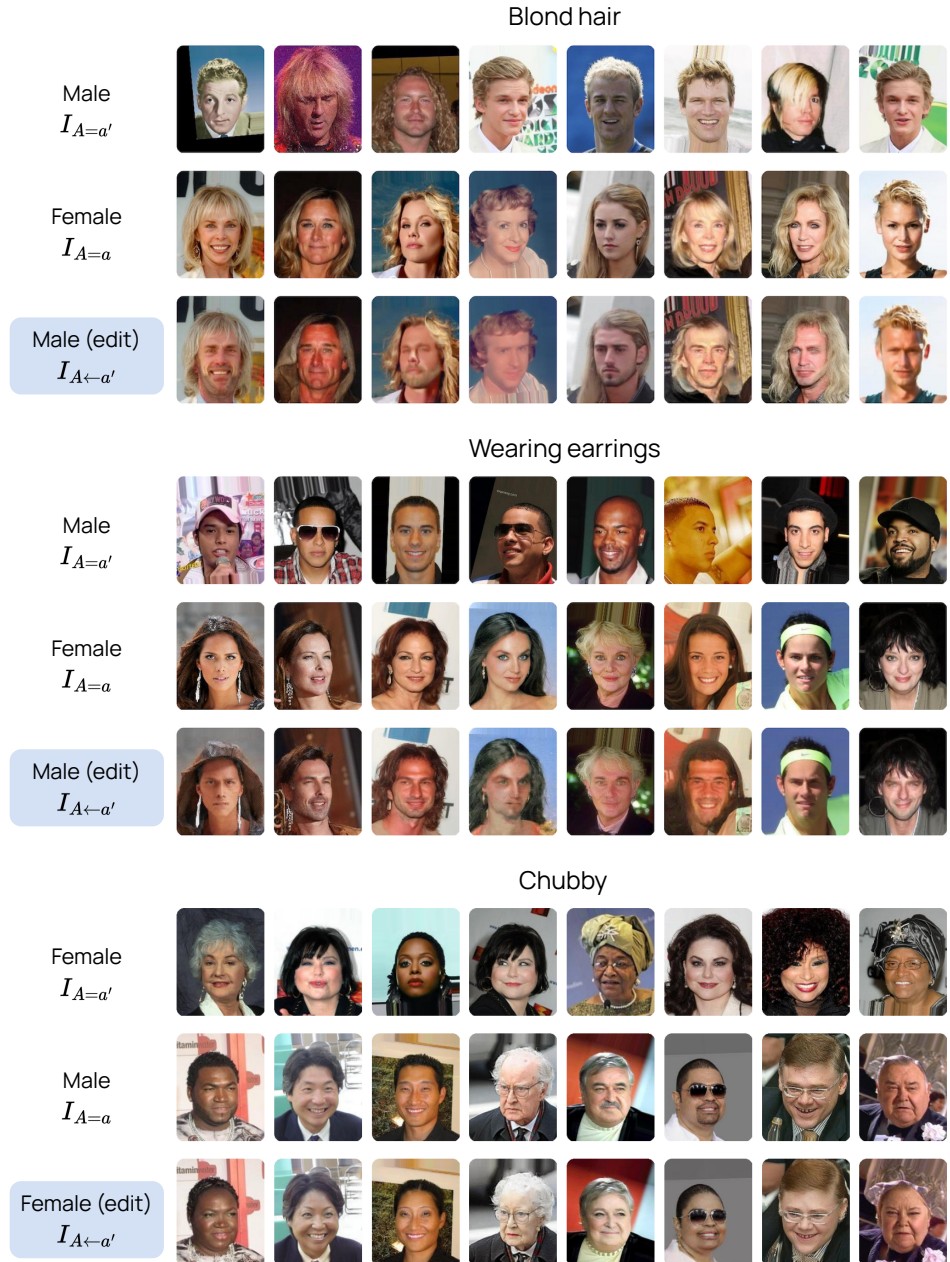

Figure 8: Counterfactual image editing on CelebA dataset. For each target attribute, we show original examples from the minority group $A = a'$ (**top**), majority group $A = a$ (**middle**), and counterfactually edited examples from $a$ to $a'$ (**bottom**).

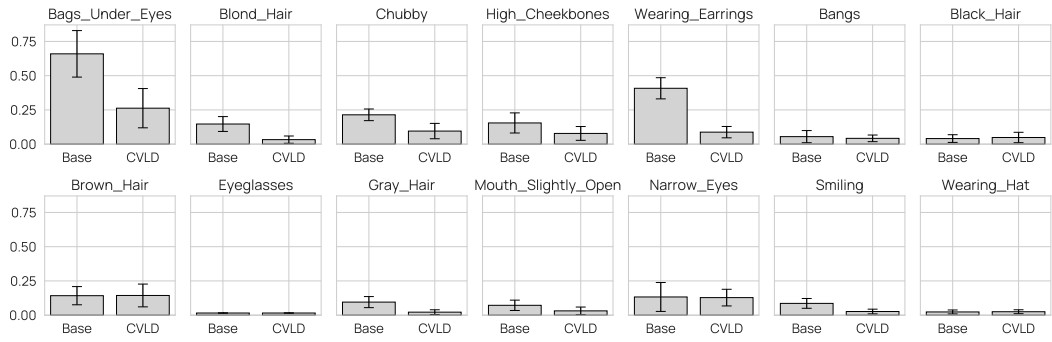

Figure 9: Per-attribute improvement over baseline fine-tuning on CelebA.

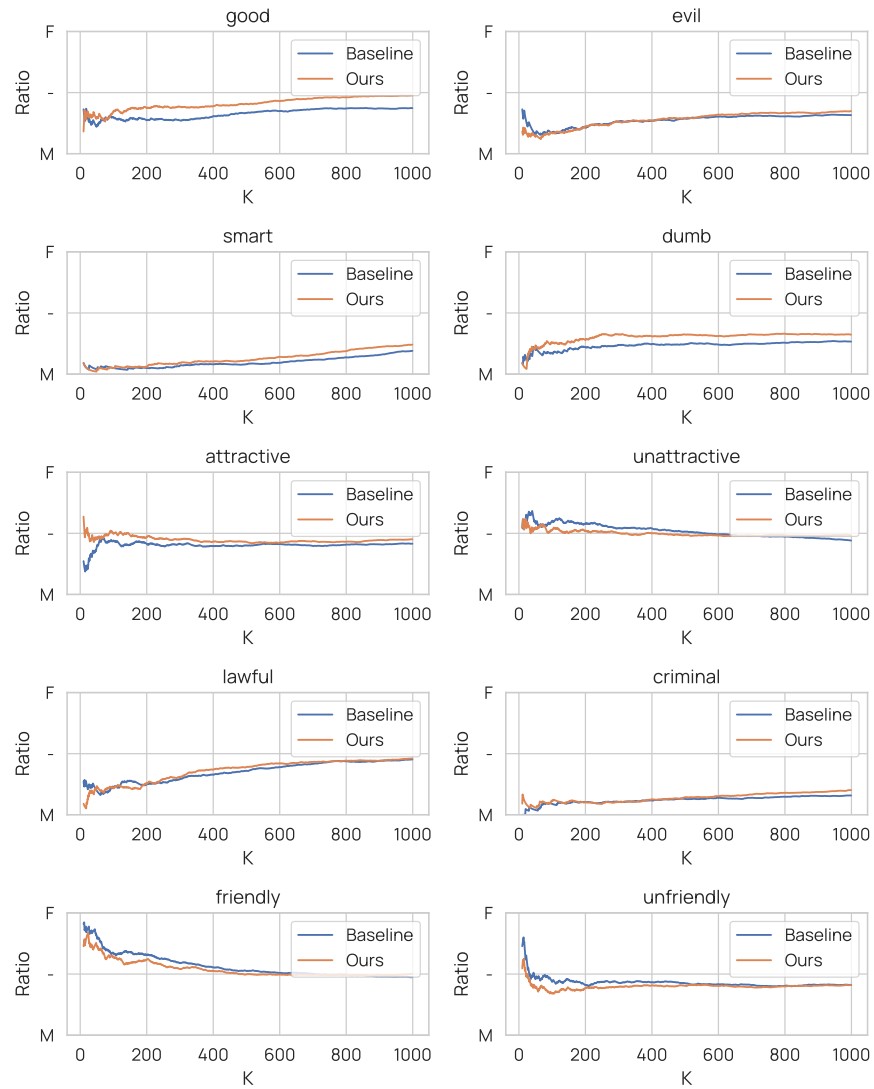

Figure 10: Gender ratio of top $K$ retrieved examples per query on FairFace.

foundation models themselves remains an open problem. It is tempting to apply **CVLD** to debias the pretraining process of foundation models, but the scale of the experiment is prohibitive due to the large number of counterfactual examples required. Finally, simply introducing interventions on protected attributes may not be sufficient to remove biases from the model. For example, the model may learn to rely on other attributes that are correlated with the protected attribute to make predictions, such as the association between "earrings" and "blond hair", even when intervention is performed on the gender attribute. We leave the study of such implicit biases to future work.

