# OpenReview forum: "Debias your VLM with Counterfactuals: A Unified Approach"
_ICLR.cc/2024/Conference — Submitted to ICLR 2024_

### Official Review · Reviewer_nQv5 · 2023-10-28

**Soundness:** 3 good
**Presentation:** 3 good
**Contribution:** 3 good
**Rating:** 5
**Confidence:** 4

**Summary:**

Vision-language models (VLMs) have achieved impressive performance on various tasks but have been shown to exhibit biases due to biased training data. In this study, the authors propose a simple debiasing framework, counterfactual vision-language debiasing
(CVLD), that aims to quantify and mitigate biases in vision-language models. CVLD introduces a causal intervention module to generate counterfactual image-text pairs and use causal fairness metrics to measure the difference in model predictions between original and counterfactual distributions. The authors also propose bias-free adaptation techniques to minimize bias in pre-trained models, achieving promising results in image classification, retrieval, and captioning tasks.

**Strengths:**

1. The paper provides a robust framework that scales to different visual-language downstream tasks like image classification, image retrieval, and image captioning tasks.

2. The proficiency of CVLD is demonstrated in a set of fine-tuning experiments across different tasks using well-established fairness measures.

3. The paper is well-written and details the objectives and results for each of the downstream tasks separately.

**Weaknesses:**

1. One of the primary weaknesses of the paper is its novelty in terms of the main framework. In order to infuse fairness, Agarwal et al. [1] introduced a triplet-based objective that maximizes the agreement between the original graph and its counterfactual views. Given that CVLD follows suit and incorporates a similar framework, the novelty is limited.

2. In most cases, the counterfactual image seems noisy and is non-reflective of the counterfactual protected attribute (e.g., in Fig. 3, we don't observe women riding the rowboat). In such cases, are the counterfactual just some noisy version of the original image? How do we attribute the debiasing to a protected attribute if the quality of the counterfactuals is not good?

3. The framework is a data-extensive approach, i.e., for debiasing, it needs a counterfactual version of each image-text pair and expensive fine-tuning of VLMs for debiasing.


**References**

1. Agarwal, C., Lakkaraju, H. and Zitnik, M. Towards a unified framework for fair and stable graph representation learning. In UAI, 2021.

**Questions:**

Please see the weaknesses for more details.

---

> ### Author Response · Authors · 2023-11-21
>
> Thank you for your insightful review and suggestions. Please refer to the general response for issues not discussed below.
>
> ---
>
> ## Novelty in terms of the main framework.
> Please see point “**Novelty**” in general comments. We appreciate the additional reference of NIFTY (Agarwal et al., 2021) which is conceptually similar and will acknowledge the work in the updated version, although we argue that the scope of the two works differ substantially: while NIFTY aims to fair and robust GNNs with a *known* underlying causal graph, our work involves vision-language transformer models on raw images and text as inputs, with the underlying graph *unknown* and simulated by image-text editing models. The primary contribution of CVLD is that it demonstrates the effectiveness of image-text editing models in debiasing VLMs across multiple downstream tasks with no architectural changes, which has not been demonstrated in prior work to our knowledge.
>
> ## Counterfactual image seems noisy.
> Refer to “**Quality of counterfactual editing**” in general response. More examples of intervened images are provided in Fig. 7 and 8, highlighting the capabilities of editing models. Quantitative measures of edit quality are also included. We agree that the noise and imperfections in the counterfactual edits are unavoidable, but would like to emphasize that CVLD has shown its effectiveness for improving model fairness on standard metrics across classification, retrieval and captioning. Such improvements cannot be achieved simply by adding random noise to training images.
>
> ## Data-extensive approach.
> See point “**Computational cost/overhead of CVLD**” in general response. It should also be noted that the experiments are deliberately conducted in a low-resource regime where we considered few-shot classification and fine-tuned retrieval/captioning on as low as 1% of COCO, which greatly reduces the cost from adapting to the full training set. CVLD is found to be effective across different scales of training set size.

---

### Official Review · Reviewer_KRWy · 2023-10-31

**Soundness:** 2 fair
**Presentation:** 3 good
**Contribution:** 3 good
**Rating:** 5
**Confidence:** 4

**Summary:**

This work considers the problem of bias mitigation on vision-language models (VLMs). The authors introduced Counterfactual vision-language debiasing (CVLD), a technique that can be summarized in two main contributions: 1- a data generation pipeline based on off-the-shelf generative models to create counterfactual augmentations from real data; 2- a bias mitigation strategy based on fine-tuning a VLM using the generate counterfactuals. The authors evaluate the proposed approach empirically on 3 tasks: image classification, image-text retrieval, and image captioning. Experiments are carried out using a model from the BLIP family and considering both fairness and task performance metrics. Overall, results show that the proposed yields the best trade-off between improving fairness (i.e. mitigating biases) and attaining good performance in the task.

**Strengths:**

- The paper tackles a critical problem and very relevant open research question: how to mitigate bias on foundation models;

- The manuscript is overall well-written and most sections are easy to follow;

- The counterfactual augmentation approach is grounded in formal definitions from the counterfactual fairness literature;

- The experimental evaluation is extensive in the number of considered tasks.

**Weaknesses:**

- One of the central claims of the work is that the proposed approach is a unified way to mitigate bias in VLMs across multiple tasks, as per the following evidence:
  - In the title (Debias your VLM with Counterfactuals: A **Unified** Approach, bold text by myself).
  - Also throughout text: e.g. In Section 2 "[...] we focus on a task-agnostic fairness framework for VLMs, unifying the study of bias across different tasks and domains." ).

   Claiming that the proposed approach is unified and task-agnostic seemed reasonable until Section 4. However, after reading through the details of how fine-tuning with synthetic data should be carried out for the three considered tasks, it seems to me that CVLD practical instantiation takes a very different format from task to task, rendering it a specific and not-unified framework for debiasing.



- One of the key parts of the introduced approach is the counterfactual data generation. However, the authors did not mention at any part of the manuscript details of the evaluation of the data generation pipeline. Moreover, it is not clear how the quality of generated counterfactuals could affect the performance of CVLD. Moreover, other fine-grained aspects such as *how much synthetic data is needed* and how the number of synthetic samples used at training time affects performance were not addressed in the manuscript, making it difficult to judge to what extent this framework would generalize to other scenarios where it might be difficult to generate high quality counterfactuals.

- Some parts of the text do not seem to reflect the actual insights that can be extracted from the results. For example, in the introduction, the authors mentioned that CVLD "demonstrates striking effectiveness for the most studied problems in the bias literature" while it is not clear from the results that the CVLD demonstrates **striking** effectiveness, neither there are references to support the statement that the considered problems in this work are the most studied ones in the bias literature.

- The experimental results are a bit confusing and hard to parse. The employed metrics in the evaluation are not standard in the literature and it is not clear whether an improvement is observed when a metric increases or decreases. Moreover, it is not clear why some methods were grouped in different parts of the tables (e.g. in Table 1 it is not clear why both CVLDs are in different sections from BLIP-PT and the ResNet-50, aren't they directly comparable?). On a similar note, it is not clear how the different bolded numbers in the tables represent and how they should be compared against each other.

**Questions:**

- How can all the three different approaches to fine-tune VLMs with counterfactual data be seen as a unified framework? Also, how would this generalize to other tasks such as, for example, counting and object detection?

- How is CVLD performance affected by counterfactual data generation quality? How did the authors assess the quality of generated data in order to know whether it was "good enough" to be employed for bias mitigation?

- How computationally expensive is the data generation approach? How does it compare to techniques that do not rely on data generation for bias mitigation?

- How is the performance of CVLD affected by the choice of the lambda hyperparameter?

---

> ### Author Response · Authors · 2023-11-21
>
> Thank you for your insightful review and suggestions. Please refer to the general response for issues not discussed below.
>
> ---
>
> ## Unified approach.
> We acknowledge that the framing of CVLD as “universal” and “task-agnostic” may have been misleading, since they refer to the data debiasing process, while the subsequent model adaptation still involves “a simple regularized training objective and minimal modifications to the adaptation pipeline” for each task, as mentioned in the Intro section. However, in defense of the title of the paper, we argue that CVLD, as a data-centric approach, has the distinctive advantage over model-centric mitigation methods in that it greatly reduces the effort to tailor the training pipeline to downstream tasks. The adaptation procedures in Sec. 4 can be distilled into two modes: For classification and captioning, the training is merely augmented by counterfactual examples with a weight parameter ($\lambda$), with no change to model computation at all. For retrieval, the contrastive loss itself is modified to account for counterfactual candidates in each batch, which reduces to simply modifying the optimization targets during training. The internals of the VLM is again unchanged, as illustrated in Fig. 3, meaning no overhead in deployment over a vanilla VLM (more discussion in “Computational cost/overhead of CVLD” in general comments). In contrast, model debiasing approaches tend to utilize additional modules, multiple training stages, and/or sophisticated loss functions, making them less trivial to adapt to a wide range of tasks. While there was not enough time to conduct the experiments, we believe that edited datasets produced by CVLD may also benefit tasks like detection/segmentation on COCO, as we find the edits to preserve the scene layout and background objects with high probability (Fig. 7 in appendix). In general, we believe that while not truly universal, CVLD is a significant step forward in minimizing the efforts needed to debias foundation VLMs on different downstream tasks.
>
> ## Evaluation of the data generation pipeline & relation to debiasing performance.
> Refer “**Quality of counterfactual editing**” in general comments. We appreciate the suggestion to include additional evaluation metrics for the counterfactual intervention. It is difficult to conclude how the edit quality metrics relate to the eventual debiasing performance, as the metrics rely on pre-trained CLIP which itself may be biased.
> However, the current version of CVLD performs sample selection based on visual similarity to the input image, and we observed in preliminary experiments that this strategy produces slightly better debiasing results than uniformly sampling edits at random, suggesting that poor quality images deviating from the input data may be detrimental.
>
> In general, we believe that the ability of CVLD is bounded by the capabilities of the editing models, and that future study may be needed to establish a definitive link between edit quality and debiasing performance. However, as demonstrated in “Evaluation beyond gender bias” in general comments, we observe positive sign that the existing editing models are powerful enough for many commonly studied forms of bias in vision-language.
>
> ## Experimental results.
> We would like to clarify that all evaluation datasets and metrics in this work are well-established standards in the literature. These include DEO for classification on CelebA (Ramaswamy et al., 2021), Bias and MaxSkew for retrieval on FairFace (Berg et al., 2022), BiasAmp and LIC for captioning on COCO (Hirota et al., 2022). For all fairness metrics lower is better, as is now indicated in the main tables of the paper. The fact that CVLD not only reduces model bias by large margins without sacrificing in-distribution performance, but also achieves so through simple image-text editing and no task-specific model debiasing, is particularly striking to us.
>
> In Tab. 1, we showed results of CVLD in both 16- and 256-shot settings, and compared them to naive BLIP using the same number of shots. In the first section of the table, BLIP-PT is not adapted to CelebA dataset while BLIP and RN-50 are fine-tuned on the full dataset, making them not directly comparable to the few-shot results below. Best results of each section of the table are highlighted in bold.
>
> ## Computational cost of data generation.
> See “**Computational cost/overhead of CVLD**” in general response.

---

> > ### Author Response · Authors · 2023-11-21
> >
> > ## Choice of the lambda hyperparameter.
> > For classification on CelebA, an ablation on choices of $\lambda$ is included in Fig. 4, and relevant discussions in Sec. 5.2. We include below the ablation results on captioning and retrieval tasks, both using COCO 1%.
> >
> > - Retrieval
> >   | $\lambda$      | 0 (baseline) | 0.1  | 0.3  | 1    |
> >   | --------------- | ------------ | ---- | ---- | ---- |
> >   | MaxSkew$^{128}$ ↓ | .281         | .224 | .222 | .257 |
> >   | Rec@1 ↑         | 59.7         | 59.5 | 59.4 | 59.2 |
> >
> >
> > - Captioning
> >   | $\lambda$ | 0 (baseline) | 0.1   | 0.3    | 1      |
> >   | ---------- | ------------ | ----- | ------ | ------ |
> >   | BiasAmp ↓  | 0.91         | 0.25  | \-0.42 | \-1.18 |
> >   | CIDEr ↑    | 130.0        | 130.5 | 130.4  | 128.8  |
> >
> > It can be seen that $\lambda = 0.3$ achieves a nice balance between fairness and performance for retrieval and captioning.

---

> > > ### Comment · Reviewer_KRWy · 2023-12-04
> > > **Response to rebuttal**
> > >
> > > Dear authors, thank you for your work in the rebuttal.
> > >
> > > While the new results related to the effect of lambda and the quality of generated data address some of my concerns, I still think claiming the proposed approach is unified is too strong. As per the comments made by the authors in the rebuttal, I agree the proposed approach is more general, but not unified (see my comments in the review). Moreover, in light of the new results regarding the quality of generated data, visual inspection of the images added to Appendix D suggests some of the counterfactuals are very low quality. Finally, some of my main concerns are still unclear, namely: how the quality of generated data impacts the final debiasing performance and how the computational cost of the proposed approach **quantitatively** compares to debiasing methods that do not require generating data. Given that, I decided to keep my original score.

---

### Official Review · Reviewer_Kjn2 · 2023-10-31

**Soundness:** 4 excellent
**Presentation:** 3 good
**Contribution:** 2 fair
**Rating:** 5
**Confidence:** 4

**Summary:**

This paper attempts to unify the study of biases across vision-language problems. It attempts to create counterfactuals to swap the gender in an image-text pair using readily available tools like LLMs and image-editing methodologies. With the help of the generated counterfactuals, it is shown that the model performances improve for multiple tasks like image retrieval, image classification and image captioning.

**Strengths:**

1. The use of LLMs and other models like Instruct Pix2Pix is smart to generate counterfactuals.
2. The authors adopt different ways to incorporate these counterfactuals into multiple downstream tasks as it is not always possible to alter the pretraining itself.
2. Using existing VLMs on top of the original datasets along with the counterfactuals seem to help reducing the bias while also maintaining model performance.

**Weaknesses:**

1. Lack of novelty: The paper simply uses some state-of-the-art LLM to generate counterfactual text, and null text inversion/InstructPix2Pix to generate the counterfactual images.
2. The paper only covers gender biases - no experiments on other biases like racial/age. Biases may exist even in non-social cases (like the water-land bias in the popular Waterbirds dataset). This has not been explored.
3. No comparison with other debiasing VLM methods ([1], [2]).
4. The paper advocates generating counterfactuals for bias mitigation. However, not many sample examples are shown even in the supplementary.
5. Not all biases (for example, models are seen to learn various spurious correlations like camels can only be present in deserts, airplanes can only be in the sky, etc) are quantifiable like gender. Is generating counterfactuals the solutions for those kinds of biases too?

[1] Zhu et al., Debiased Fine-Tuning for Vision-Language Models by Prompt Regularization, AAAI 2023
[2] Chuang et al., Debiasing Vision-Language Models via Biased Prompts, arxiv 2023

**Questions:**

1. The method of generating counterfactuals does not generate diverse images, but only modifies the existing images. Generating diverse images can help the models further. Can this be addressed?
2. What if multiple biases are present at once? Like gender and race together. Can this method of generating counterfactuals handle such scenarios?

---

> ### Author Response · Authors · 2023-11-21
>
> Thank you for your insightful review and suggestions. Please refer to the general response for issues not discussed below.
>
> ---
>
> ## Only covers gender biases; unquantifiable biases & multiple biases.
> Please see “**Evaluation beyond gender bias**” in general response. Additional results on *background* bias on Waterbirds and *racial* bias on COCO are included, demonstrating that CVLD is generally applicable to various forms of biases. Our understanding is that CVLD is potentially useful for all settings where 1) bias can be described in natural language, and 2) corresponding causal interventions (e.g., change gender/race of subjects, or replace the background) can be reliably performed by the image-text editing models.
>
> Simultaneous mitigation of multiple biases is a great suggestion. One simple way is to extend CVLD with multiple edits, one for each protected attribute, which presents a challenge to the compositionality of editing models. However, as we intend for CVLD to become a baseline in data-centric VLM debiasing, we believe that the study of multiple bias attributes lies beyond the scope of this work and should be left for follow-up research.
>
> ## Lack of novelty; comparison to other methods.
> Please see point “**Novelty**” in general comments. CVLD showcases the effectiveness of image-text editing models in debiasing VLMs across multiple downstream tasks with no architectural changes, which has not been demonstrated in prior work. We consider the simplicity of the proposed framework a strength, as it does not require tweaking the internal model architecture or sophisticated training procedures, making the method more practical for tuning and deploying models at scale. It also makes the method complementary to model debiasing approaches, which may further benefit from the availability of debiased data.
>
> ## Qualitative results.
> As mentioned in “**Quality of counterfactual editing**” in general response, we have added qualitative results of generated images to the appendix of the paper, as well as a set of quantitative evaluation for edit quality.
>
> ## Generating diverse examples.
> This is a great point. While the editing models are not deterministic in their output, text-to-image generation certainly may improve the diversity of the images, albeit at a cost of lower accuracy. Using the CelebA dataset we studied the potential of generating examples of target attributes (e.g., “a person with blond hair”) with Stable Diffusion, instead of editing existing images. This has the benefit of scaling easily to larger numbers of examples in theory, yet in our experiments, the trained models on generated samples (BLIP-SD) underperform those trained with real/edited data substantially, as shown in the table. The gap only widens after increasing the number of shots from 16 to 256. This can be explained by a larger domain gap between generated images and real images, compared to a much smaller gap with editing models.
>
> | Model            | DEO ↓ | ΔAcc ↓ | AP ↑ | mAcc ↑ |
> | ---------------- | ----- | ------ | ---- | ------ |
> | BLIP 256-shot    | 16.0  | 14.2   | 88.7 | 82.8   |
> | &nbsp;&nbsp; \+ CVLD          | 10.2  | 11.2   | 89.4 | 83.9   |
> | BLIP-SD 256-shot | 23.9  | 39.7   | 78.3 | 65.5   |
> | &nbsp;&nbsp; \+ Unbiased      | 15.1  | 25.8   | 78.3 | 68.5   |
> | BLIP 16-shot     | 20.1  | 20.7   | 82.3 | 76.5   |
> | &nbsp;&nbsp; \+ CVLD          | 15.6  | 18.2   | 82.2 | 76.5   |
> | BLIP-SD 16-shot  | 16.8  | 27.8   | 77.8 | 65.1   |
> | &nbsp;&nbsp; \+ Unbiased      | 11.5  | 31.0   | 77.6 | 67.4   |
>
> Anecdotally, we also found the diffusion models to struggle on certain attribute combinations of CelebA, such as men wearing earrings. Editing models on the other hand produce more accurate results by visual inspection. Further study may be needed to understand the trade-off between accuracy and diversity of synthetic images.

---

### Official Review · Reviewer_sFrN · 2023-11-04

**Soundness:** 2 fair
**Presentation:** 3 good
**Contribution:** 3 good
**Rating:** 5
**Confidence:** 3

**Summary:**

This paper proposes a simple framework to debias vision-language models. First, one generates a text prompt for the target image that can guide the image editing procedure. Second, one generates a counterfactual image from the text prompt that has been edited by flipping the bias-related word (e.g., boy -> girl). Third, one fine-tunes the target VLM with the generated counterfactual images. The empirical results show that this method can be used to mitigate the gender bias of many VLMs.

**Strengths:**

- **Soundness.** The proposed framework is very reasonably designed; it makes perfect sense that such a method will work, given access to well-performing LLMs and text-based image editors.

- **Novelty.** As far as I know, the method CVLD is novel.

- **Significance of the topic.** VLMs are now one of the core backbones of most machine learning applications, and thus having a safety guarantee on such foundation models is a very important yet understudied topic.

- **Writing.** The paper is clearly written and easy to read, despite having many typos.

**Weaknesses:**

- **Limited Empirical Evaluation.** The proposed method has been evaluated almost exclusively on a specific type of bias---the gender bias. This is a very severe limitation for a paper which frames itself as targeting general bias in VLMs; the paper exemplifies racial bias multiple times in the text. If the authors are exclusively targeting the gender bias, a significant portion of this paper should be re-written to clarify this point.

- **Relies on external models, which may be prone to other types of bias.** The debiasing procedure of this paper relies on the generative/editing capabilities of existing models (e.g., prompt-to-prompt editing). This is a vulnerability in terms of a bias, because such edited images may be prone to other types of biases that may be difficult to detect (see, e.g., Bias-to-Text by Kim et al. (2023)). I wonder if authors could demonstrate any "robustness" of the proposed paradigm to the potential biases hidden in the LLMs or prompts.

- **(minor) Clarity.** Figure 1 is not very informative and difficult to parse what the figure is trying to say. What the sketch part is trying to say is unclear (perhaps more details in the caption will be better). Also, it took me some time to notice that "M -> F" means male -> female. The "lock" figures are somewhat difficult to tell whether they are locked or unlocked (maybe use frozen <-> fire analogy, like many other papers, or use additional color cues?).

**Questions:**

Please see the "weaknesses" section.

---

> ### Author Response · Authors · 2023-11-21
>
> Thank you for your insightful review and suggestions. Please refer to the general response for issues not discussed below.
>
> ---
>
> ## Limited empirical evaluation.
> Please see “**Evaluation beyond gender bias**” in general response. In short, while we experimented primarily with gender bias, CVLD is by no means exclusive to gender editing, and is as capable as the ability of the image-text editing models to alter protected attributes under text prompt. We show additional results on 1) background bias on Waterbirds, and 2) racial bias on COCO, to demonstrate the potential of the method.
>
> ## Bias from editing models.
> Thanks for the insightful suggestion. As discussed in “**Quality of counterfactual editing**” in general comments, it is true that the external editing models may be biased themselves, although very few studies have attempted to study this bias in a quantitative manner. Therefore, we resort to standard fairness benchmarks in classification, retrieval and captioning in the main paper for evaluating the debiasing quality. The fact that CVLD reduces model bias across dataset and tasks can be seen as strong evidence that the potential bias of external models is not enough to offset their effectiveness to remove spurious bias in the training data.
>
> Although it is difficult to directly study the robustness of CVLD to the bias of LLMs/diffusion models, there are steps taken to reduce the potential impact. For example, by sampling multiple edits per image and keeping those most similar to the original, we minimize the chance of inadvertently altering image regions not causally related to the protected attribute (e.g., if some edits cause the color of the clothes or background objects to be changed while others preserve the original colors, the latter would be accepted).
>
> ## Improve clarity of figures.
> Thanks for the suggestion. We have revised Fig. 1 and 3 to improve the clarity of modules and editing operations. We will continue to improve the readability of the paper.

---

### Author Response · Authors · 2023-11-21
**General Response**

We would like to thank all reviewers for their insightful comments and suggestions. We provide responses to common questions below, and further address the concerns of each reviewer in the review thread.

*R1 = `sFrN`, R2 = `Kjn2`, R3 = `KRWy`, R4 = `nQv5`*

---

## Evaluation beyond gender bias (R1, R2)

It should be noted that the proposed CVLD framework is not limited to gender biases: for any intervention procedure that the editing model is capable of, it is possible to create counterfactual examples for debiased training. The empirical evaluation of the paper focused on gender, since gender bias is prevalent across datasets and tasks in vision-language, and ground-truth annotations are relatively easier to obtain than race or gender. We will update the manuscript to reflect the focus of the work. Still, to show the generality of CVLD, we experiment with two settings beyond gender bias:

- **Image classification with background bias.** We use the WaterBirds dataset [1], using water/land background as protected attribute $A$; an intervention on $A$ therefore requires replacing the background of the original images, and can be realized through editing prompts like “move the bird to a [ocean/lake/land/forest]”. The fairness of trained models are measured by worst group accuracy following prior works, though we also include the unweighted average of group accuracies as a measure of model performance on an unbiased test set.

  *Example: https://ibb.co/k50gQcF*

  |                | Average acc. ↑ | AP ↑   | Worst group acc. ↑ | Mean group acc. ↑ |
  | -------------- | ------------ | ---- | ---------------- | --------------- |
  | RN50 (FT)      | 97.3         | –    | 60.0             | –               |
  | &nbsp;&nbsp; \+ DRO [1]    | 97.4         | –    | 76.9             | –               |
  | BLIP (FT)      | 97.5         | 88.7 | 75.5             | 87.6            |
  | &nbsp;&nbsp; \+ CVLD        | 97.3         | 90.9 | 81.9             | 89.6            |
  | BLIP (64-shot) | 94.0         | 62.7 | 26.3             | 71.4            |
  | &nbsp;&nbsp; \+ CVLD        | 92.9         | 78.2 | 60.8             | 81.0            |

  \* *Following [1], average accuracy is weighted by group priors in the biased training set.*

  When fine-tuning the BLIP model on the full dataset, the naive BLIP model roughly matches the performance and fairness of a ResNet-50 trained with group DRO, while our proposed debiasing further boosts the worst group accuracy by 6%. Under a low-shot setting, the models become even more susceptible to the background bias as the worst group accuracy drops significantly from full fine-tuning. CVLD increases worst group accuracy by more than a two-fold, suggesting a greater advantage of counterfactual debiasing when dataset size is limited.

- **Image captioning with racial bias**. We use the COCO dataset and perform gender interventions on all images with human subjects. Since the ground-truth racial group is unknown, we generate one counterfactual for each target group $a’ \in \mathcal{A}$, regardless of the input image. We then adapt the BLIP models on the intervened images, and compare their fairness on the test set through the LIC metric, using racial annotations by [2].

  *Example: https://ibb.co/L1Q812H*

  |                | LIC ↓  | BLEU4 ↑ | CIDEr ↑ |
  | -------------- | ------ | ------- | ------- |
  | BLIP PT        | \-0.60 | 38.5    | 129.2   |
  | BLIP (COCO 1%) | 1.12   | 39.0    | 130.0   |
  | &nbsp;&nbsp; \+ CVLD        | \-2.12 | 39.3    | 130.4   |

  As the table shows, CVLD produces significantly lower leakage (LIC) for race than both the pre-trained BLIP and standard fine-tuning, while preserving high captioning scores measured by BLEU4 and CIDEr.

[1] Sagawa et al. Distributionally Robust Neural Networks for Group Shifts: On the Importance of Regularization for Worst-Case Generalization. https://arxiv.org/abs/1911.08731

[2] Zhao et al. Understanding and Evaluating Racial Biases in Image Captioning. https://arxiv.org/abs/2106.08503

---

> ### Author Response · Authors · 2023-11-21
>
> ## Quality of counterfactual editing (R2, R3, R4)
>
> We have revised the paper to include qualitative examples of image-text editing on COCO and CelebA datasets in the appendix (Fig. 7 and 8). On both datasets, the models produced reasonable gender interventions while preserving the overall visual context.
>
> We also include below a quantitative evaluation of the intervention quality:
> - **CLIP score** [3] measures the overall consistency between synthetic images and target captions.
> - **Gender accuracy** specifically evaluates the success rate of attribute intervention through a CLIP gender classifier (using “a photo of a man/woman” as prompts).
> - **Directional CLIP similarity** [4] measures the affinity between *shifts* in edited images and text captions in the CLIP embedding space.
> - **Visual similarity** captures the ability to retain visual features from the input images apart from the bias attribute.
>
> Evaluation on COCO reveals that InstructPix2Pix produces more visually consistent images to the original image (higher visual similarity), while null-text inversion alters the image more substantially leading to higher directional similarity and gender accuracy. Notably both the CLIP score and gender accuracy are lower for edited images than original ones. This could be due to the counterfactual edits producing scenes underrepresented in the CLIP training data, in addition to visual artifacts in synthetic images.
>
> | *all scores in %*                    | CLIP score ↑ | Gender acc. ↑ | Direction sim. ↑ | Visual sim. ↑ |
> | ------------------- | ------------ | ------------- | ---------------- | ------------- |
> | Original            | 31.3         | 82.2          | —                | —             |
> | InstructPix2Pix     | 29.7         | 64.3          | 14.7             | 91.3          |
> | Null-text Inversion | 29.9         | 67.8          | 21.1             | 87.8          |
>
> It should be noted that the evaluation itself relies on pretrained models like CLIP which can themselves be biased: higher edit quality by the above metrics do not guarantee lower bias in the edited data. Therefore, we resort to standard fairness benchmarks in classification, retrieval and captioning in the main paper for evaluating the debiasing quality. An extended version of the analysis between edit quality and debiasing performance will be added to the paper upon acceptance.
>
> [3] Hessel et al. CLIPScore: A Reference-free Evaluation Metric for Image Captioning. https://arxiv.org/abs/2104.08718
>
> [4] Gal et al. StyleGAN-NADA: CLIP-Guided Domain Adaptation of Image Generators. https://arxiv.org/abs/2108.00946
>
>
> ## Novelty (R2, R4)
>
> We concede that the proposed solution is straightforward, with minimal modification to the models other than a few task-specific losses. However, we argue that the greater contribution of the work is that it demonstrates the effectiveness of image-text editing diffusion models in debiasing VLMs across multiple downstream tasks with no architectural changes, which has not been demonstrated in prior work to our knowledge. This makes possible for CVLD to be applied to novel datasets and tasks as long as the intervention on the protected attribute can be performed, as exemplified in the background and racial bias experiments above. Furthermore, as CVLD performs debiasing on the dataset level, it can be considered a complementary approach to the vast majority of existing mitigation methods for model debiasing, instead of a competing one. We will continue to explore the possibility of combining data and model debiasing in future work.
>
> ## Computational cost/overhead of CVLD (R3, R4)
>
> The complexity of CVLD depends on the type of the image editing models. Of the two used in this work, InstructPix2Pix is more efficient as it uses a standard generation pipeline of Stable Diffusion, unlike null-text inversion which involves hundreds of iterations to optimize. In practice, using a batch size of 8 images, we found the editing time for Instruct/Null-text editing to be around 10/30 seconds on a single RTX3090 card. While this may less efficient than model-centric approaches for a *single* training episode, we note that data debiasing does not need to be repeated for *retraining* models (e.g., for hyperparameter search), which is typically the most time consuming phase of a research or product development cycle. Once the counterfactual data is generated, VLM debiasing reduces to training on an augmented dataset, with no additional overhead to both training and inference.

---

### Meta-Review · Area_Chair_JTmm · 2023-12-09

**Metareview:**

This paper aims to address biases across vision-language problems by generating counterfactuals. The positive feedback appreciates the attempt to unify bias mitigation strategies, particularly using state-of-the-art Language Models (LLMs) and models like Instruct Pix2Pix for generating counterfactuals. The paper explores multiple downstream tasks and demonstrates improved model performance in image retrieval, image classification, and image captioning. However, critical feedback highlights limitations and gaps, such as the exclusive focus on gender biases, neglecting other biases like racial/age and biases in non-social datasets. The absence of comparisons with other debiasing Vision-Language Models (VLMs), a lack of sufficient examples demonstrating bias mitigation through generated counterfactuals, and unclear insights drawn from experimental results contribute to the reviewers' skepticism. Additionally, concerns are raised about the non-quantifiable nature of certain biases and the impact of data quality on debiasing performance. The paper received borderline reject scores from all reviewers and the authors' rebuttal did not change their stand. Also, the counterfactual images are criticized for being noisy and non-reflective of the intended protected attribute, contributing to the overall recommendation for rejection.

**Justification For Why Not Higher Score:**

All the reviewers were not positive and the rebuttal did not change their opinion.

**Justification For Why Not Lower Score:**

N/A

---

### Decision · Program_Chairs · 2024-01-16

Reject